# Physiology-Informed Diffusion for 12-Lead ECG Generation

## Abstract

Large-scale 12-lead ECG data are critical for training reliable cardiac machine learning systems, yet their availability is limited by privacy constraints, annotation cost, and severe class imbalance. Generative models offer a promising solution, but standard diffusion models typically treat ECGs as generic multivariate time series and do not explicitly exploit known physiological structure.

We propose PhysDiff-ECG, a physiology-guided diffusion framework that integrates cardiac ordinary differential equation (ODE) prior into the diffusion trajectory. Our central idea is to make ECG physiology tractable during training by deriving differentiable regularizers from a dynamical model of cardiac activity together with a differentiable 12-lead observation model. Given a denoised reconstruction along the reverse process, PhysDiff-ECG fits a latent physiological explanation via an unrolled inner optimization and penalizes violations of both the simulator dynamics and the induced ECG reconstruction.

This training-time regularization biases the learned denoising trajectories toward physiologically realizable ECGs while preserving the flexibility of latent diffusion. Experiments on standard 12-lead ECG benchmarks show that PhysDiff-ECG improves physiological fidelity, representation-space realism, class-conditional diagnostic consistency, and downstream classification performance relative to strong GAN and diffusion baselines.

## 1 Introduction

Electrocardiography (ECG) is a fundamental diagnostic modality for assessing cardiac function, with the standard 12-lead ECG providing a rich, multi-view measurement of the heart's electrophysiological activity. Large-scale, diverse ECG datasets are critical for training reliable machine learning systems for tasks such as arrhythmia detection, risk stratification, and clinical decision support. However, access to such data is often severely bottlenecked by privacy concerns, regulatory constraints, and demographic or pathological imbalances (Voigt & Bussche, 2017). These challenges have motivated a growing interest in generative models for synthesizing realistic 12-lead ECG signals that can augment training data and support simulation studies without compromising patient privacy (de Melo et al., 2022; Giuffrè & Shung, 2023).

Generative models, particularly Generative Adversarial Networks (GANs) (Goodfellow et al., 2014) and, more recently, Denoising Diffusion Probabilistic Models (DDPMs) (Ho et al., 2020), have achieved strong results in realistic data synthesis. Diffusion models in particular provide stable training, improved mode coverage, and high-quality samples by reversing a gradual noising process, and have become competitive across images, audio, and sequential data. However, applying standard diffusion models directly to physiological signals exposes a key mismatch. Unlike natural images, whose generative manifold is largely defined by surface statistics, ECG recordings are observations of a structured physical system: cardiac electrophysiology and its body-surface projection (McSharry et al., 2003; Potse, 2018). As a result, treating ECGs as generic multivariate waveforms can yield samples that match local morphology yet violate global constraints, since waveform timing and inter-lead relationships are governed by well-characterized biophysical dynamics and measurement geometry. In this work, by global physiological constraints we refer to properties that cannot be verified from local waveform realism in a single lead alone. These include coordinated beat timing across leads, physiologically plausible waveform morphology, and consistency among the 12 leads as correlated observations of shared cardiac electrical activity. Thus, a generated ECG may appear realistic lead-by-lead

while still being implausible as a 12-lead recording, for example if R-peaks are temporally misaligned across leads or if different leads imply incompatible cardiac activation patterns. Appendix Fig. 3 illustrates typical unconstrained generation failures with physiologically consistent 12-lead structure.

Standard data-driven generative models are largely agnostic to these governing laws. As a result, they may reproduce local ECG morphology while failing to respect the latent dynamical structure and cross-lead relationships induced by cardiac electrophysiology and body-surface projection. In the ECG domain, however, we do have access to tractable physiological models that relate latent cardiac dynamics to observed 12-lead signals. A natural question, then, is how to incorporate such known structure into modern generative models in a way that is differentiable, trainable, and useful for improving the realism and downstream utility of generated ECGs.

Physiological structure in ECG generation is not itself new, and prior work has explored constraints based on lead relationships, simulator-based priors, and physiology-aware diffusion models. Our contribution is therefore not the generic use of physiology, but a specific training-time formulation that combines latent diffusion, differentiable fitting of a low-dimensional physiological explanation, and regularization through both dynamical and observation-model consistency.

In this work, we propose *PhysDiff-ECG*, a diffusion-based framework that bridges data-driven generative modeling and dynamic-based cardiac electrophysiology. Rather than treating physiology as a post-processing constraint or a separate simulator, we incorporate it directly into diffusion training through a differentiable regularization mechanism. Specifically, given a denoised reconstruction along the reverse process, we fit a low-dimensional physiological explanation using a reduced dynamical model together with a differentiable 12-lead observation model, and penalize violations of both the latent dynamics and the reconstructed ECG. In this way, physiology acts as a structured inductive bias on the learned denoising dynamics while preserving the flexibility and scalability of latent diffusion.

Crucially, this physiology-informed regularization provides a structured training signal during denoising. By encouraging denoised reconstructions to admit a plausible physiological explanation, our approach improves biological validity while also improving representation-space fidelity, diagnostic consistency, and downstream usefulness of the generated samples.

Our contributions are threefold:

- **Physiology-Informed Latent Diffusion for 12-Lead ECGs.** We introduce PhysDiff-ECG, a physiology-informed latent diffusion framework that incorporates tractable physiological structure through differentiable regularization derived from a reduced cardiac dynamical model and a differentiable 12-lead observation model.
- **Improved Fidelity and Downstream Utility.** We show that enforcing physiological structure improves physiological fidelity, representation-space realism, class-conditional diagnostic alignment, and downstream ECG classification performance relative to strong GAN and diffusion baselines.
- **Training Stability.** We show that physiology-informed regularization improves sample quality and reduces metric variability across training iterations, relative to the same diffusion backbone without the physiological regularizer.

## 2 Related Work

The emergence of deep generative models shifted ECG synthesis toward data-driven approaches, with Generative Adversarial Networks (GANs) playing a central role in early work. Methods such as WaveGAN (Donahue et al., 2018) and Pulse2Pulse (Thambawita et al., 2021) demonstrated the feasibility of generating realistic ECG waveforms using adversarial training.

Subsequent work has explored a range of approaches for multi-lead ECG generation, spanning both single-beat synthesis and full 12-lead signal modeling. Vector-quantized variational autoencoders (VQ-VAEs) have been used to learn compact ECG representations (Liu et al., 2020), while 3KG (Gopal et al., 2021) performs data augmentation in vectorcardiogram (VCG) space using three-dimensional geometric transformations. ME-GAN (Chen et al., 2022) extends GAN-based architectures to multi-view ECG synthesis, and (Huang

et al., 2023) incorporates unsupervised noise modeling to improve robustness. More recently, MultiODE-GAN (Yehuda & Radinsky, 2024) combines mechanistic ODE formulations with adversarial learning for multi-lead ECG generation, building on classical dynamical models (McSharry et al., 2003). These approaches improve realism and inter-lead consistency, but many remain tied to single-beat generation.

More recently, diffusion-based generative models have emerged as a strong alternative. Denoising Diffusion Probabilistic Models (DDPMs) (Ho et al., 2020) have achieved strong results in time-series modeling (Rasul et al., 2021; Kollovieh et al., 2023) and have been adapted to biomedical signals, including ECGs. Methods such as SSSD-ECG (Alcaraz & Strodthoff, 2023) apply diffusion modeling to multi-lead ECG generation, while DiffuSETS (Lai et al., 2025) conditions ECG synthesis on auxiliary information such as diagnostic labels, clinical text, or patient-specific metadata.

Despite their empirical success, many existing GAN and diffusion based ECG generators still treat ECGs primarily as generic multivariate time series and rely mainly on statistical structure learned from data. In contrast, earlier biophysical approaches based on dipole models or reaction-diffusion systems (Potse, 2018; Quiroz-Juárez et al., 2019) explicitly encode cardiac electrophysiology, but are not designed to capture the full variability, noise characteristics, and pathological diversity of real-world ECG recordings. This leaves a gap between flexible data-driven generators and physiologically structured simulators.

Incorporating domain knowledge into generative models has also been explored through physics-informed learning and constrained deep generative approaches. Within physics-informed deep learning, Physics-Informed Neural Networks (PINNs) embed differential-equation residuals directly into training objectives, primarily for forward simulation and inverse parameter estimation rather than generative modeling (Raissi et al., 2019; Karniadakis et al., 2021). In cardiac electrophysiology, differentiable ODE solvers have been used to infer patient-specific parameters for ODE/PDE-based models such as FitzHugh-Nagumo (Boulakia et al., 2010), enabling physiologically interpretable reconstruction and personalization (Sayadi et al., 2010; Cantwell et al., 2019). However, these methods are not designed to generate diverse, high-dimensional signals such as full 12-lead ECGs.

Our work is most closely related to physiology-aware diffusion for ECG generation, but differs in how physiological structure is incorporated. Rather than treating physiology as a post-processing constraint or a sampling-time correction, we integrate it into diffusion training through a differentiable regularization mechanism. Given a denoised reconstruction, we fit a low-dimensional physiological explanation using a reduced cardiac dynamical model together with a differentiable 12-lead observation model, and penalize violations of both the latent dynamics and the induced ECG reconstruction.

# 3 Method

We propose PhysDiff-ECG, a latent diffusion framework for class-conditioned 12-lead electrocardiogram (ECG) generation regularized by a differentiable physiological prior. Our central idea is to make ECG physiology tractable during training by combining a low-dimensional dynamical model of cardiac activity with a differentiable observation model that maps latent physiological states to 12-lead surface ECGs.

Given a full ECG signal, a pretrained encoder maps the signal to a latent representation on which diffusion is performed. The denoiser is trained with the standard diffusion objective together with a physiology-informed regularization term evaluated on denoised reconstructions along the reverse process. This regularizer encourages intermediate reconstructions to admit a plausible physiological explanation, thereby steering the learned denoising dynamics toward physiologically realizable ECGs.

## 3.1 Problem Setup

Let $S \in \mathbb{R}^{12 \times T}$ denote a 12-lead ECG segment with $T$ samples and class label $y$. In diffusion notation, we write $S_0$ for the clean signal, where the subscript refers to diffusion time rather than ECG sample index.

We assume that $S$ admits an approximate low-dimensional physiological explanation through a latent trajectory $X(\tau) \in \mathbb{R}^d$, evolving in simulator time $\tau$, and a differentiable observation model $G(\cdot; \theta_{\text{obs}})$ that

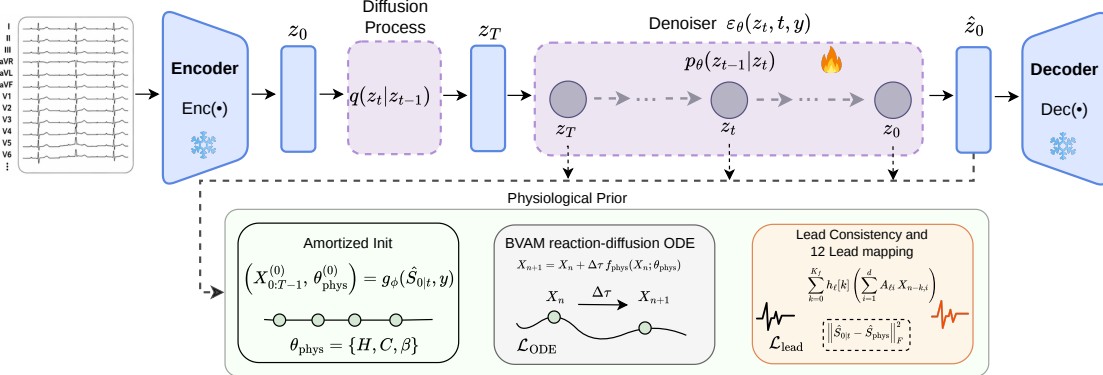

Figure 1: Overview of the physiology-informed latent diffusion framework for 12-lead ECG generation.

maps latent physiological states to the 12-lead surface ECG:

$$\frac{dX(\tau)}{d\tau} = f_{\text{phys}}(X(\tau); \theta_{\text{phys}}),$$

$$S^{\text{phys}}(\tau) = G(X(\tau); \theta_{\text{obs}}).$$

Here, $f_{\text{phys}}$ is the reduced cardiac dynamical model, $\theta_{\text{phys}}$ denotes simulator parameters inferred per reconstruction, and $\theta_{\text{obs}}$ denotes shared observation-model parameters.

We discretize the simulator on $N$ steps of size $\Delta\tau$, writing $X_n \approx X(n\Delta\tau)$ for $n = 0, \ldots, N-1$. To compare simulator outputs with the observed ECG, we align the simulator grid with the signal length by setting (or resampling to) $N = T$. Throughout, simulator time $\tau$ is distinct from diffusion step $t$: $\tau$ indexes the latent physiological trajectory within a single ECG, whereas $t$ indexes the denoising process.

Throughout the paper, $t$ denotes diffusion time, $\tau$ denotes simulator time, and $n$ denotes the discrete ECG/sample index. Thus, $z_t$ is the noisy latent variable at diffusion step $t$, whereas $X_n \approx X(n\Delta\tau)$ is the physiological latent state at simulator step $n$. The clean ECG is denoted by $S_0$, and the denoised reconstruction estimated from $z_t$ is denoted by $\hat{S}_{0|t}$.

### 3.2 Latent Diffusion Preliminaries

We perform diffusion in the latent space of a pretrained VAE. Given a clean ECG signal $S_0$, the encoder produces a latent code $z_0 = \text{Enc}(S_0)$, and the decoder reconstructs the signal as $\tilde{S}_0 = \text{Dec}(z_0)$. Unless otherwise stated, the VAE is pretrained and kept fixed while training the diffusion model.

For a variance-preserving diffusion process with schedule $\{\alpha_t\}_{t=1}^K$ and $\bar{\alpha}_t = \prod_{s=1}^t \alpha_s$, the forward process is

$$q(z_t \mid z_0) = \mathcal{N}\big(\sqrt{\bar{\alpha}_t}\, z_0,\, (1 - \bar{\alpha}_t)\mathbf{I}\big). \tag{1}$$

We train a class-conditioned denoiser $\epsilon_\theta(z_t, t, y)$ using the standard noise-prediction objective

$$\mathcal{L}_{\text{diff}} = \mathbb{E}_{z_0, t, \varepsilon \sim \mathcal{N}(0, I)} \left[ \|\varepsilon - \epsilon_\theta(z_t, t, y)\|_2^2 \right]. \tag{2}$$

Given a noisy latent state $z_t$, we form the corresponding denoised estimate

$$\hat{z}_{0|t} = \frac{1}{\sqrt{\bar{\alpha}_t}} \left( z_t - \sqrt{1 - \bar{\alpha}_t}\, \epsilon_\theta(z_t, t, y) \right), \qquad \hat{S}_{0|t} = \text{Dec}(\hat{z}_{0|t}). \tag{3}$$

The decoded signal $\hat{S}_{0|t}$ serves as the input to the physiology-informed regularizer introduced in the next subsection.

### 3.3 Physiological Prior: Reaction–Diffusion System

The first component of the physiological regularizer is a dynamical constraint on the latent cardiac trajectory. Given a denoised reconstruction $\hat{S}_{0|t}$, we infer a discrete latent trajectory $X_{0:T-1}$ that should both explain the reconstructed ECG and remain close to a reduced cardiac dynamical model. We use the BVAM reaction–diffusion ODE as this reduced model, and its vector field $f_{\mathrm{phys}}$ is later used to define the ODE-consistency penalty in Section 3.5. As the physiological prior, we use the low-dimensional ODE obtained by spatial discretization of the BVAM reaction–diffusion system of Quiroz-Juárez et al. (2019). This reduced nonlinear system serves as a compact surrogate model of cardiac conduction dynamics. With $X = [x_1, \ldots, x_d]^\top \in \mathbb{R}^d$ (specifically $d = 6$ for this system), the dynamics are

$$\dot{x}_1 = x_1 - x_2 - Cx_1x_2 - x_1x_2^2, \tag{4}$$

$$\dot{x}_2 = Hx_1 - 3x_2 + Cx_1x_2 + x_1x_2^2 + \beta\,(x_4 - x_2), \tag{5}$$

$$\dot{x}_3 = x_3 - x_4 - Cx_3x_4 - x_3x_4^2, \tag{6}$$

$$\dot{x}_4 = Hx_3 - 3x_4 + Cx_3x_4 + x_3x_4^2 + \beta\,(x_6 - 2x_4 + x_2), \tag{7}$$

$$\dot{x}_5 = x_5 - x_6 - Cx_5x_6 - x_5x_6^2, \tag{8}$$

$$\dot{x}_6 = Hx_5 - 3x_6 + Cx_5x_6 + x_5x_6^2 + \beta\,(x_4 - x_6), \tag{9}$$

where $\theta_{\mathrm{phys}} = \{H, C, \beta\}$. Following Quiroz-Juárez et al. (2019), $H$ controls the oscillatory regime and effective heart rate, $C$ controls the nonlinearity of the restitution dynamics, and $\beta$ determines the coupling strength induced by the discrete Laplacian.

The BVAM prior comes from a spatially discretized reaction-diffusion model of cardiac electrical activity and provides a compact system of coupled nonlinear oscillators for generating ECG-like dynamics. We do not use it as a full cellular, anatomical, or patient-specific electrophysiology simulator. In particular, it does not model detailed cellular ionic dynamics, tissue anisotropy, torso geometry, electrode placement variability, or patient-specific conduction.

We use BVAM as a differentiable surrogate that captures coarse dynamical properties relevant to ECG generation, including rhythmic excitation, nonlinear restitution-like behavior, and coupling between latent physiological components. The ODE-consistency term is therefore applied as a soft regularizer rather than a hard constraint, allowing the diffusion model to retain data-driven flexibility when the reduced prior is imperfect.

We treat $\theta_{\mathrm{phys}}$ as latent variables inferred from the denoised reconstruction. Class information enters the overall model through the class-conditioned denoiser $\epsilon_\theta(z_t, t, y)$ and may also be provided to the initializer $g_\phi$.

**Numerical integration.** To evaluate the ODE-based physiology regularizer, we approximate the continuous latent dynamics with an explicit Euler discretization. Given the current latent state $X_n$, the next state is approximated as

$$X_{n+1} = X_n + \Delta\tau\, f_{\mathrm{phys}}(X_n; \theta_{\mathrm{phys}}), \quad n = 0, \ldots, N - 2. \tag{10}$$

This first-order solver provides a simple and differentiable approximation of the ODE trajectory, allowing the regularizer to penalize deviations from the prescribed physiological dynamics. Although higher-order solvers could be used, Euler integration keeps the computation lightweight and the residual easy to optimize.

### 3.4 Differentiable 12-Lead Mapping

A limitation of the original BVAM formulation is that it produces only a single ECG lead through a scalar linear combination of the latent states, $\mathrm{ECG}(t) = \sum_{i=1}^d \alpha_i x_i$. To generate full 12-lead ECGs, we replace this scalar observation model with a differentiable multi-lead mapping.

In the simplest case, we use a shared linear mixing matrix $A \in \mathbb{R}^{12 \times d}$:

$$\hat{S}_{\mathrm{phys}}[\ell, n] = \sum_{i=1}^d A_{\ell i}\, X_{n,i}, \tag{11}$$

where $\ell \in \{1, \ldots, 12\}$ and $n \in \{0, \ldots, T-1\}$. This model reduces to the original single-lead formulation when selecting one row of $A$. The matrix $A$ is shared across samples and learned jointly with the rest of the model.

This linear observation model serves as a flexible approximation to the ECG lead-field operator: each lead is modeled as a distinct projection of a shared low-dimensional latent trajectory. Accordingly, the 12 leads are treated as correlated views of common underlying cardiac dynamics rather than as independent channels.

To capture small lead-specific temporal delays and smoothing effects, we optionally augment the observation model with a short causal FIR filter $h_\ell \in \mathbb{R}^{K_f + 1}$ for each lead:

$$\hat{S}_{\text{phys}}[\ell, n] = \sum_{k=0}^{K_f} h_\ell[k] \left( \sum_{i=1}^{d} A_{\ell i} X_{n-k, i} \right). \tag{12}$$

For out-of-range indices with $n - k < 0$, we use boundary handling via $X_{n-k} = X_0$ (or zero-padding). With this extension, the full observation parameters are $\theta_{\text{obs}} = \{A, \{h_\ell\}_{\ell=1}^{12}\}$, shared across samples and learned jointly with the rest of the model. Setting $h_\ell[k] = \mathbb{1}[k = 0]$ recovers the purely linear model in equation 11.

### 3.5 Physiology-Informed Regularization

Given a denoised reconstruction $\hat{S}_{0|t} \in \mathbb{R}^{12 \times T}$, we define a physiology-informed regularizer by fitting a latent physiological trajectory that explains the reconstruction through the simulator and observation model.

We initialize this inner fitting problem using an amortized network $g_\phi$:

$$\left( X_{0:T-1}^{(0)}, \theta_{\text{phys}}^{(0)} \right) = g_\phi(\hat{S}_{0|t}, y).$$

The network predicts an initial latent trajectory $X_{0:T-1}^{(0)}$ and simulator parameters $\theta_{\text{phys}}^{(0)}$, where $(0)$ denotes initialization before refinement. We then apply $M$ unrolled gradient steps to obtain $(\hat{X}_{0:T-1}, \hat{\theta}_{\text{phys}})$, yielding a differentiable simulator-consistent explanation of $\hat{S}_{0|t}$.

**ODE consistency.** We encourage the inferred latent trajectory to follow the simulator dynamics through the Euler residual

$$\mathcal{L}_{\text{ODE}} = \sum_{n=0}^{T-2} \left\| \hat{X}_{n+1} - \left( \hat{X}_n + \Delta\tau \, f_{\text{phys}}(\hat{X}_n; \hat{\theta}_{\text{phys}}) \right) \right\|_2^2. \tag{13}$$

**Lead consistency.** Let $\hat{S}_{\text{phys}} = \mathcal{G}(X; \theta_{\text{obs}})$ denote the ECG reconstructed from the inferred latent trajectory using equation 11 or equation 12. We define the signal-consistency term over all 12 leads as

$$\mathcal{L}_{\text{lead}} = \left\| \hat{S}_{0|t} - \hat{S}_{\text{phys}} \right\|_F^2. \tag{14}$$

This encourages the denoised reconstruction to be globally consistent with the physiological observation model across all leads.

**Inner physiology objective.** We combine the two terms into

$$\mathcal{L}_{\text{inner}} = \lambda_{\text{ODE}} \, \mathcal{L}_{\text{ODE}} + \lambda_{\text{lead}} \, \mathcal{L}_{\text{lead}}. \tag{15}$$

In practice, $(\hat{X}_{0:T-1}, \hat{\theta}_{\text{phys}})$ are obtained by approximately solving

$$(\hat{X}_{0:T-1}, \hat{\theta}_{\text{phys}}) \approx \arg \min_{X_{0:T-1}, \theta_{\text{phys}}} \mathcal{L}_{\text{inner}}(\hat{S}_{0|t}; X_{0:T-1}, \theta_{\text{phys}}, y) \tag{16}$$

This inner optimization induces a reconstruction-dependent physiological explanation

$$(\hat{X}_{0:T-1}(\hat{S}_{0|t}), \hat{\theta}_{\text{phys}}(\hat{S}_{0|t})).$$

The outer physiology regularizer used during diffusion training is then defined as

$$\mathcal{L}_{\text{phys}}(\hat{S}_{0|t}) := \mathcal{L}_{\text{inner}}\big(\hat{S}_{0|t}; \hat{X}_{0:T-1}(\hat{S}_{0|t}), \hat{\theta}_{\text{phys}}(\hat{S}_{0|t}), y\big).$$

Because the inner optimization is unrolled, $\mathcal{L}_{\text{phys}}(\hat{S}_{0|t})$ is differentiable with respect to $\hat{S}_{0|t}$ and can be backpropagated through the denoising network during training.

In practice, we use a small fixed number of unrolled refinement steps ($M \ll T$), which keeps the additional computational overhead moderate relative to the diffusion backbone.

### 3.6 Training Objective and Reverse-Process Coupling

For each sampled diffusion timestep, the denoiser is trained to predict the diffusion noise while its implied clean reconstruction is regularized to admit a plausible physiological explanation under the simulator and observation model (see Fig. 1).

Given a noisy latent $z_t$, the denoiser estimates $\hat{z}_{0|t}$ via equation 3, which is decoded into $\hat{S}_{0|t} = \text{Dec}(\hat{z}_{0|t})$. We then infer a latent physiological explanation $(\hat{X}, \hat{\theta}_{\text{phys}})$ for $\hat{S}_{0|t}$ using the unrolled optimization in equation 16, and evaluate the regularization loss $\mathcal{L}_{\text{phys}}(\hat{S}_{0|t})$. This loss is added to the diffusion objective and backpropagated through the denoiser, thereby coupling latent denoising with physiological consistency.

The regularized denoising objective at timestep $t$ is

$$\ell_t(\theta) = \|\varepsilon - \epsilon_\theta(z_t, t, y)\|_2^2 + \lambda_{\text{phys}} \, \mathcal{L}_{\text{phys}}(\hat{S}_{0|t}), \tag{17}$$

where $\hat{S}_{0|t} = \text{Dec}(\hat{z}_{0|t})$ and $\hat{z}_{0|t}$ is defined by equation 3; the dependence of $\mathcal{L}_{\text{phys}}$ on the inner unrolled refinement is implicit.

Averaging over training samples, diffusion timesteps, and Gaussian noise yields the full objective

$$\mathcal{L}_{\text{total}} = \mathbb{E}_{S_0, y, t, \varepsilon} \left[ \|\varepsilon - \epsilon_\theta(z_t, t, y)\|_2^2 + \lambda_{\text{phys}} \, \mathcal{L}_{\text{phys}}(\hat{S}_{0|t}) \right], \quad z_0 = \text{Enc}(S_0), \ z_t = \sqrt{\bar{\alpha}_t} \, z_0 + \sqrt{1 - \bar{\alpha}_t} \, \varepsilon. \tag{18}$$

This objective is backpropagated through the chain

$$z_t \mapsto \hat{z}_{0|t} \mapsto \hat{S}_{0|t} \mapsto (\hat{X}, \hat{\theta}_{\text{phys}}),$$

thereby encouraging the denoiser to produce reverse-process trajectories whose denoised reconstructions are consistent with both the latent cardiac dynamics and the differentiable 12-lead observation model.

We optimize over the full latent trajectory $X_{0:T-1}$ rather than only an initial condition in order to keep the inner problem flexible and numerically stable when fitting noisy intermediate reconstructions. The ODE residual therefore acts as a soft dynamical constraint, encouraging—but not strictly forcing—the refined trajectory to remain close to the simulator manifold.

Importantly, $\mathcal{L}_{\text{phys}}$ is not introduced as a post-hoc sampling correction, but as a training-time regularizer on the learned denoising dynamics. At inference time, sampling proceeds using the learned denoiser, which has already been biased toward physiologically realizable trajectories during training.

Unlike label-conditioning terms that inject class information into the reverse process, our physiological term does not encode semantic class preferences. Instead, it acts as a structured model-based regularizer derived from a low-dimensional cardiac dynamics prior and a differentiable multi-lead observation model, penalizing denoised reconstructions that fail to admit a plausible physiological explanation.

In each training iteration, we sample an ECG $S_0$ and label $y$, encode it to $z_0$, sample a diffusion timestep $t$ and noise $\varepsilon$, construct $z_t$ via equation 1, predict $\hat{z}_{0|t}$ and decode $\hat{S}_{0|t}$, run $M$ unrolled refinement steps to obtain $(\hat{X}, \hat{\theta}_{\text{phys}})$, and optimize equation 18.

## 4 Experimental Setup

### 4.1 Model Architecture

PhysDiff-ECG builds on the latent diffusion architecture of DiffuSETS (Lai et al., 2025) and adapts it to class-conditional 12-lead ECG generation. The model consists of a pretrained VAE and a class-conditioned denoising diffusion model operating in the VAE latent space. Given an ECG $\mathbf{x} \in \mathbb{R}^{12 \times T}$, the encoder maps it to a latent code $\mathbf{z}_0 = \text{Enc}(\mathbf{x}) \in \mathbb{R}^{C \times L}$, and the decoder reconstructs $\hat{\mathbf{x}} = \text{Dec}(\mathbf{z}_0)$. In our implementation, we use $C = 6$ latent channels and latent length $L = 128$. Additional architectural details are provided in Appendix B.

### 4.2 ECG Datasets

We evaluate PhysDiff-ECG on two standard 12-lead ECG benchmarks, using one as the primary development dataset and the other for external validation. Our primary experiments are conducted on PTB-XL (Wagner et al., 2020), a large-scale benchmark for 12-lead ECG analysis. It contains 21,799 recordings from 18,869 patients, with each recording spanning 10 seconds and annotated with diagnostic labels. We use PTB-XL as the main dataset for training the generative model and for downstream synthetic-data evaluation. For additional validation, we also evaluate on the Georgia 12-Lead ECG Challenge dataset (G12EC) (Alday et al., 2020). It contains 10,344 recordings from 7,871 patients, each 10 seconds long and sampled at 500 Hz. We use G12EC to test whether the conclusions observed on PTB-XL generalize to a second large-scale 12-lead ECG cohort.

### 4.3 Evaluation Protocol

**Data Splits and Cross-Validation.**  To prevent data leakage, we use patient-wise splits throughout. For each dataset, 20% of patients are held out for final testing (Xu & Goodacre, 2018), and the remaining 80% form the development set, within which we perform 5-fold cross-validation. Additional preprocessing and split details are provided in Appendix B.

### 4.4 Downstream Classifiers for Synthetic-Data Evaluation

To assess the utility of synthetic ECGs generated by PhysDiff-ECG, we evaluate downstream classification performance using two strong 12-lead ECG classifiers: a ResNet-based model (Ribeiro et al., 2020) and an attention-augmented ResNet (Nejedly et al., 2021). Using two distinct classifier families reduces the chance that improvements from synthetic augmentation are specific to a single downstream architecture. Architectural details are provided in Appendix B.

## 5 Experimental Results

We evaluate PhysDiff-ECG along three complementary axes: (i) signal- and feature-level fidelity, (ii) class-conditional diagnostic alignment, and (iii) downstream classification performance under synthetic-data augmentation.

### 5.1 Compared Generative Models

We compare PhysDiff-ECG against the following generative baselines:

- **WaveGAN** (Donahue et al., 2018): Originally proposed for raw audio synthesis, WaveGAN models temporal dependencies in one-dimensional signals and is adapted here for multi-lead ECG waveform generation.
- **Pulse2Pulse** (Thambawita et al., 2021): A conditional GAN employing a U-Net style encoder-decoder with one-dimensional convolutions for ECG waveform generation.
- **SSSD-ECG** (Alcaraz & Strodthoff, 2023): A diffusion-based ECG generator that leverages structured state-space (S4) layers within a DDPM framework to model long-range temporal dependencies.

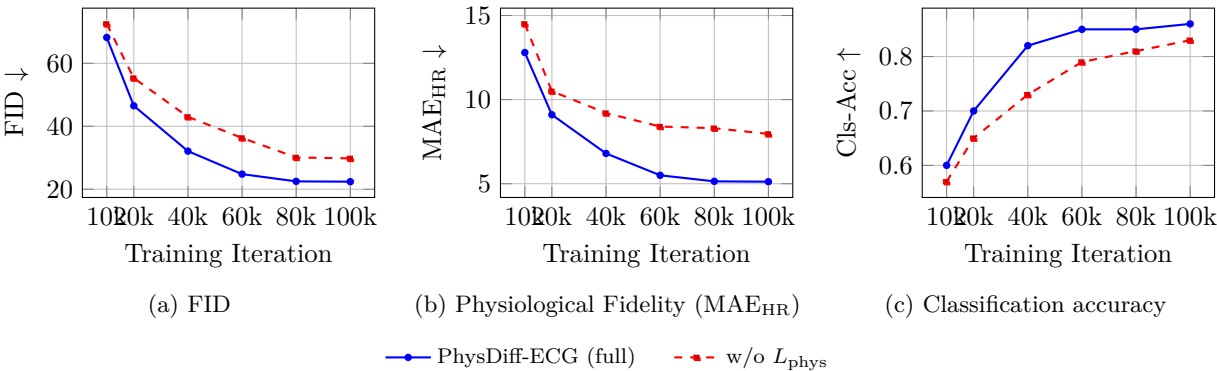

Figure 2: Training-Iterations evaluation of ECG generation quality. We compare the full PhysDiff-ECG framework with the same diffusion backbone trained without the physiology-informed regularizer. We report FID ,heart-rate mean absolute error $\text{MAE}_{\text{HR}}$ and classifier accuracy.

- **DiffuSETS** (Lai et al., 2025): A diffusion framework originally designed for text-to-ECG synthesis using LLM–derived semantic embeddings, adapted here to a class-conditioned generation.
- **Diffusion-TS** (Yuan & Qiao, 2024): A diffusion model for multivariate time-series generation based on encoder-decoder transformer with disentangled temporal representations. Supports both conditional and unconditional generation.
- **PhysDiff-ECG**: Our class-conditioned latent diffusion model with physiology-informed regularization, which incorporates a cardiac ODE prior.

## 5.2 Signal and Feature-Level Evaluation

**Physiological Fidelity (Heart-Rate Preservation).** To assess whether generated ECGs preserve basic physiological characteristics, we compare the heart rate (HR) extracted from generated samples $\hat{S}$ to that of matched real samples from the same class. We report the mean absolute error $\text{MAE}_{\text{HR}}$ in beats per minute (bpm), where lower values indicate better HR preservation. Heart rate is computed using a standard R-peak detector and converting the median RR interval to bpm.

**Signal- and Feature-Level Distribution Similarity.** We assess distributional realism using Fréchet distance in a learned ECG feature space. Real and generated ECGs are embedded with a fixed pretrained Net1D encoder, and the Fréchet distance between the resulting Gaussian fits is computed as

$$\text{FID} = \|\mu_r - \mu_g\|_2^2 + \text{Tr}\left(\Sigma_r + \Sigma_g - 2\left(\Sigma_r \Sigma_g\right)^{1/2}\right),$$

where $(\mu_r, \Sigma_r)$ and $(\mu_g, \Sigma_g)$ are the empirical mean and covariance of the embedded real and generated samples. Lower FID is better.

**Class-Conditional Diagnostic Alignment.** Because PhysDiff-ECG is class-conditioned, we also test whether generated ECGs are diagnostically consistent with their conditioning label $y$. To do so, we apply a fixed pretrained ECG classifier to generated samples and report accuracy.

Table 1 summarizes $\text{MAE}_{\text{HR}}$, representation-space FID, and classifier accuracy across all generative models.

## 5.3 Downstream Classification Performance

To evaluate downstream utility, we train classifiers on either real ECGs only or the same real ECGs augmented with synthetic samples generated by each baseline and by PhysDiff-ECG, and test exclusively on unseen real ECGs (Alcaraz & Strodthoff, 2023; Yehuda & Radinsky, 2024). We report per-abnormality sensitivity,

Table 1: Evaluation of ECG generation quality across methods. We report heart-rate mean absolute error $MAE_{HR}$ in bpm, representation-space Fréchet distance (FID), and classifier accuracy (Cls-Acc). Lower is better for $MAE_{HR}$ and FID, while higher is better for Cls-Acc.

| Model | $MAE_{HR}$ ($\downarrow$) | FID ($\downarrow$) | Cls-Acc ($\uparrow$) |
|---|---|---|---|
| WaveGAN | 15.70 | 50.4 | 0.71 |
| Pulse2Pulse | 12.01 | 41.2 | 0.76 |
| SSSD-ECG | 13.82 | 39.4 | 0.77 |
| DiffuSETS (class-cond.) | 9.22 | 29.8 | 0.82 |
| Diffusion-TS | 10.18 | 34.3 | 0.80 |
| PhysDiff-ECG | **5.14** | **22.4** | **0.86** |

specificity, and AUC in a one-vs-rest setting (Wang et al., 2018; Bressman et al., 2020; Golany et al., 2022). For sensitivity and specificity, thresholds are selected on the validation split and applied to the test set.

Table 2 reports results for the ECG classifier (Ribeiro et al., 2020). The Baseline CLS column uses only real training data, whereas all other columns augment the same real data with synthetic samples generated by the corresponding model. For each abnormality, all methods use the same synthetic augmentation size, set to $N$, where $N$ is the number of real training samples for that abnormality. Augmenting with PhysDiff-ECG improves performance over the Real Only setting and is competitive with or better than alternative generative baselines on most abnormalities. Gains are typically larger for lower-prevalence conditions, where limited real data constrains generalization. Results with an additional downstream classifier are reported in Sec. C.5.

Table 2: Downstream classification performance (evaluated on a held-out **real** test set). We report per-abnormality sensitivity (Sens.), specificity (Spec.), and AUC for the baseline classifier and for classifiers trained with synthetic augmentation from each Model. Boldface indicates the best value per row within each metric.

| Abnormality | Baseline CLS* | | | Pulse2Pulse | | | SSSD-ECG | | | DiffuSETS | | | PhysDiff-ECG | | |
|---|---|---|---|---|---|---|---|---|---|---|---|---|---|---|---|
| | Sens. | Spec. | AUC | Sens. | Spec. | AUC | Sens. | Spec. | AUC | Sens. | Spec. | AUC | Sens. | Spec. | AUC |
| AFL | 0.80 | 0.86 | 0.84 | 0.81 | 0.86 | 0.85 | 0.83 | 0.88 | 0.87 | 0.83 | 0.87 | 0.87 | **0.85** | **0.90** | **0.89** |
| TAb | 0.89 | 0.70 | 0.88 | 0.90 | 0.71 | 0.89 | 0.92 | 0.72 | 0.91 | 0.93 | 0.71 | 0.91 | **0.94** | **0.73** | **0.93** |
| QAb | 0.84 | 0.73 | 0.87 | 0.84 | 0.74 | 0.87 | 0.86 | 0.74 | 0.88 | 0.85 | 0.75 | 0.89 | 0.86 | **0.77** | **0.90** |
| SA | 0.67 | 0.53 | 0.65 | 0.68 | 0.55 | 0.67 | 0.68 | 0.58 | 0.68 | 0.70 | 0.57 | 0.69 | 0.70 | **0.61** | **0.71** |
| LAD | 0.90 | 0.87 | 0.95 | 0.90 | 0.88 | 0.95 | 0.91 | 0.88 | 0.96 | 0.90 | 0.90 | 0.96 | **0.93** | 0.90 | **0.97** |
| LBBB | 0.98 | 0.97 | 0.99 | 0.98 | 0.96 | 0.99 | 0.98 | 0.97 | 0.99 | 0.98 | 0.97 | 0.99 | 0.98 | 0.97 | 0.99 |
| PAC | 0.88 | 0.62 | 0.79 | 0.88 | 0.64 | 0.80 | 0.89 | 0.64 | 0.81 | 0.88 | 0.65 | 0.81 | **0.90** | **0.67** | **0.84** |
| NSIVCB | 0.81 | 0.67 | 0.82 | 0.82 | 0.67 | 0.83 | 0.84 | 0.68 | 0.84 | 0.84 | 0.67 | 0.84 | 0.84 | **0.71** | **0.85** |

* Baseline classifier follows Ribeiro et al. (2020)

**Convergence During Training**

We track FID, $MAE_{HR}$, and class-conditional diagnostic accuracy across training epochs in order to understand how the physiology-informed regularizer affects optimization. Specifically, we compare the full model against the variant trained without physiological regularization.

Fig. 2 shows that physiology-informed regularization improves these metrics throughout training and often reduces performance variance across steps. This suggests that the physiological prior provides a stabilizing inductive bias during training, improving intermediate sample quality in addition to final performance.

Additional ablations, including loss-component analysis, evaluation on the G12EC dataset, and robustness across downstream classifiers, are reported in Appendix C. Qualitative results are provided in Appendix A.

## 6 Conclusion

We presented PhysDiff-ECG, a physiology-guided diffusion framework for class-conditioned 12-lead ECG synthesis. Rather than treating physiology as a post-processing constraint, PhysDiff-ECG incorporates a low-dimensional cardiac dynamical model and a differentiable 12-lead observation model directly into training through unrolled physiological fitting and consistency-based regularization. Empirically, this formulation improves representation-space realism, physiological fidelity, class-conditional diagnostic alignment, and downstream classification performance under synthetic-data augmentation relative to strong GAN and diffusion baselines. Our analysis further indicates that physiology-informed diffusion improves sample quality and reduces metric variability across training checkpoints, which is consistent with faster convergence.

A limitation of PhysDiff-ECG is its reliance on the informativeness of the chosen physiological prior. If the reduced ODE or observation model is severely misspecified, the regularizer may provide a weaker signal or bias generation toward an incomplete physiological manifold. Future work should explore richer and more adaptive electrophysiological priors, including patient-specific parameterizations and uncertainty-aware physiological constraints.We discuss broader impact, clinical-use limitations, privacy, bias, and responsible-use considerations in Appendix D.

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

# A    Qualitative Results

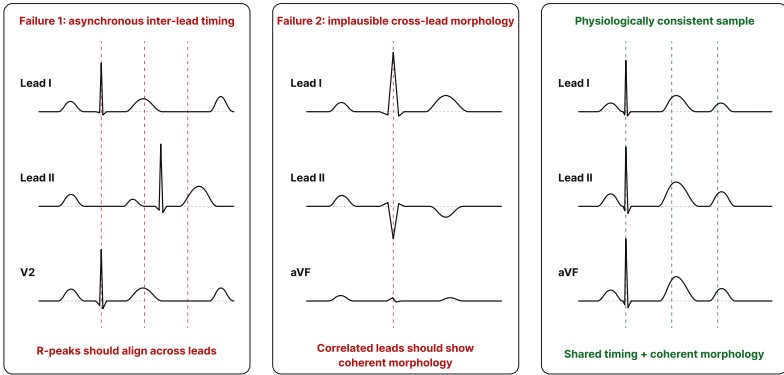

Figure 3: Illustration of global physiological constraints in 12-lead ECG generation. Unconstrained generation may produce locally plausible leads while violating synchronized inter-lead timing or coherent cross-lead morphology. PhysDiff-ECG encourages generated signals to share a consistent latent physiological trajectory.

Figure 3 illustrates two common failure modes: asynchronous R-peaks across leads and implausible morphology among correlated leads, both of which can occur even when individual leads appear locally realistic.

Figures 4–5 present two representative 12-lead ECGs generated by PhysDiff-ECG. Each figure shows a complete 12-lead recording in standard clinical order (I, II, III, aVR, aVL, aVF, V1–V6). The samples display coherent temporal rhythms across leads, realistic waveform morphology, and consistent inter-lead structure.

These qualitative examples complement the quantitative results in the main paper, demonstrating that PhysDiff-ECG yields globally coherent ECGs rather than independently plausible lead-wise signals.

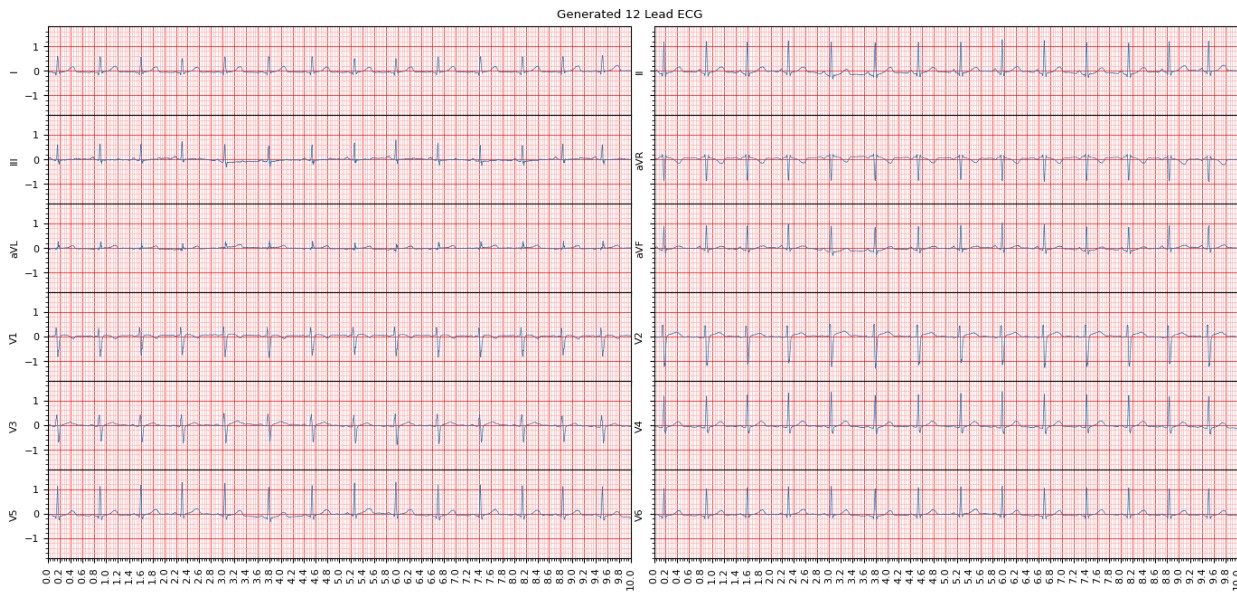

Figure 4: Generated 12-lead ECG example #1 by PhysDiff-ECG.

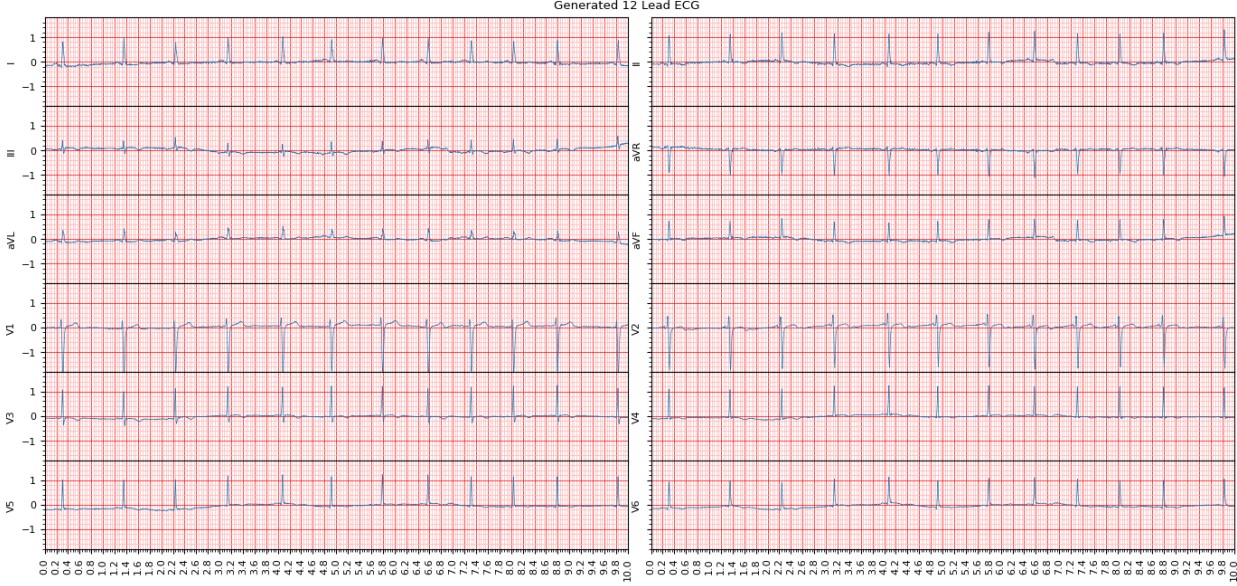

Figure 5: Generated 12-lead ECG example #2 by PhysDiff-ECG .

## B  Additional Implementation Details

**Generative model architecture.**  The denoiser $\epsilon_\theta(\mathbf{z}_t, t, y)$ is a 1D U-Net with seven resolution levels and kernel size 7, organized as a symmetric downsampling/upsampling hierarchy with skip connections. Self-attention blocks are inserted at selected resolutions to capture long-range temporal dependencies. Class conditioning is implemented through learned label embeddings injected into the denoiser by cross-attention at each resolution level. Unlike text-conditioned diffusion architectures, the conditioning signal here consists of discrete diagnostic labels.

**Data splits and cross-validation.**  For each dataset, 20% of patients are held out as an independent test set and used exclusively for final evaluation (Xu & Goodacre, 2018). The remaining 80% form the development set, within which we perform 5-fold cross-validation for model selection and variability estimation. All splits are constructed at the patient level, so that recordings from the same individual never appear in both development and test sets.

**Residual Network (ResNet).**  Our first downstream classifier is a ResNet-based architecture adapted to ECG signals (Ribeiro et al., 2020). It consists of an initial convolutional layer followed by five residual blocks with batch normalization, ReLU activations, dropout, and skip connections. Temporal downsampling is performed via strided convolutions, and a global average pooling layer feeds a final classification head.

**Attention-Enhanced ResNet.**  Our second downstream classifier augments the ResNet backbone with multi-head attention (Nejedly et al., 2021), enabling stronger modeling of long-range temporal dependencies and inter-lead interactions. This architecture has been shown to perform strongly on large-scale ECG classification benchmarks, making it a useful complementary evaluation backbone.

**PhysDiff-ECG Training and Computational Cost.**  The model is trained using the AdamW optimizer with a learning rate of $2 \times 10^{-4}$ and a batch size of 128. For the physiology-informed regularizer, we set the loss weights to $\lambda_{ODE} = 0.5$ and $\lambda_{lead} = 1.0$, determined via grid search on the validation set. Computationally, this unrolled optimization introduces approximately a 15% overhead per training iteration compared to the standard diffusion backbone. However, because $\mathcal{L}_{phys}$ is exclusively applied as a training-time regularizer, the inference cost remains strictly identical to the base diffusion model, requiring no ODE integration during sample generation.

**Baseline Adaptation and Comparison Protocol.**  We used official implementations where available. All methods were trained using the same patient-wise splits, preprocessing, diagnostic labels, and training recordings, and generated the same number of samples per class. WaveGAN was configured for 12-channel input and output, while DiffuSETS was adapted from its original conditioning setup to diagnostic-label conditioning and used the same latent dimensionality as our diffusion backbone. The remaining methods were evaluated in their native multivariate configurations. No physiological regularization was added to any baseline. Hyperparameters were selected on the same development folds, and all samples were evaluated using the same pipeline.

## C  Ablation Study

We conduct ablations to isolate the contribution of the physiology-informed regularization in PhysDiff-ECG and its constituent components. Unless otherwise stated, all ablations use the same training data, diffusion backbone, and class conditioning, and are evaluated using the same signal-/feature-level metrics (Sec. 5.2) and downstream classification protocol (Sec. 5.3).

### C.1  Ablation on Physiology-Informed Regularization

We first ablate the role of the physiology-informed regularizer during training, and then analyze the contribution of its two components: ODE consistency and lead-space consistency.

**With vs. without physiology-informed regularization.** We compare the full PhysDiff-ECG against a variant trained without the physiological regularizer, i.e., with $\lambda_{\mathrm{phys}} = 0$ in Eq. equation 18. Both variants use the same pretrained VAE, diffusion backbone, training data, and class conditioning; the only difference is whether the denoiser is trained with the additional physiological regularization term.

As shown in Table 3, removing the physiological regularizer degrades representation-space fidelity (FID), heart-rate preservation ($\mathrm{MAE_{HR}}$), and class-conditional diagnostic accuracy, indicating that the physiological prior improves both realism and label consistency.

**Regularizer components.** Our inner physiology objective combines latent ODE consistency $\mathcal{L}_{\mathrm{ODE}}$ and lead-space reconstruction $\mathcal{L}_{\mathrm{lead}}$. To isolate their roles, we train variants in which each term is removed by setting its weight to zero. Specifically, we compare the full model against variants without $\mathcal{L}_{\mathrm{ODE}}$ and without $\mathcal{L}_{\mathrm{lead}}$.

Table 3 shows that both terms contribute. Removing $\mathcal{L}_{\mathrm{ODE}}$ primarily harms heart-rate preservation, consistent with weaker enforcement of latent physiological dynamics, whereas removing $\mathcal{L}_{\mathrm{lead}}$ degrades FID and class-conditional diagnostic accuracy, consistent with weaker alignment to the observed ECG space. The full objective yields the best overall trade-off across metrics.

Table 3: Ablation of physiology-informed regularization and its components. We compare the full model against variants trained without the full physiological regularizer, without $\mathcal{L}_{\mathrm{lead}}$, and without $\mathcal{L}_{\mathrm{ODE}}$. Lower is better for FID and $\mathrm{MAE_{HR}}$; higher is better for Cls-Acc.

| Variant | $\mathbf{MAE_{HR}}$ ($\downarrow$) | **FID** ($\downarrow$) | **Cls-Acc** ($\uparrow$) |
|---|---|---|---|
| Full PhysDiff-ECG | **5.14** | **22.4** | **0.86** |
| w/o $\mathcal{L}_{\mathrm{phys}}$ ($\lambda_{\mathrm{phys}}{=}0$) | 7.96 | 27.1 | 0.84 |
| w/o $\mathcal{L}_{\mathrm{lead}}$ ($\lambda_{\mathrm{lead}}{=}0$) | 6.88 | 26.2 | 0.83 |
| w/o $\mathcal{L}_{\mathrm{ODE}}$ ($\lambda_{\mathrm{ODE}}{=}0$) | 9.10 | 28.9 | 0.81 |

These ablations also address prior dependence: removing either the dynamical residual or the observation-model consistency degrades performance, indicating that the gains are not due only to additional loss weighting but to the structured physiological signal.

### C.2 Cardiac versus Generic Dynamics Regularization

To determine whether the gains arise from cardiac-specific structure rather than generic latent-dynamics regularization, we replace the BVAM vector field with a generic neural ODE,

$$f_{\mathrm{gen}}(X; \psi) = \mathrm{MLP}_\psi(X), \tag{19}$$

where $\psi$ is shared across samples and learned jointly with the diffusion model. Its dynamics loss follows the same Euler-residual form:

$$\mathcal{L}_{\mathrm{gen}} = \sum_{n=0}^{T-2} \|X_{n+1} - (X_n + \Delta\tau f_{\mathrm{gen}}(X_n; \psi))\|_2^2. \tag{20}$$

The latent dimension, observation model, unrolled refinement of $X$, loss weights, diffusion backbone, and training protocol remain unchanged. This baseline provides nonlinear ODE-based regularization without encoding cardiac reaction–diffusion structure.

As shown in Table 4, the generic neural ODE slightly improves heart-rate error and diagnostic accuracy relative to removing the dynamics term, but degrades FID. In contrast, the BVAM prior improves all three metrics. This indicates that the gains are not explained solely by generic latent-dynamics regularization and that the cardiac structure provides an additional benefit.

Table 4: Comparison of cardiac-specific and generic latent-dynamics regularization under the same training and evaluation protocol.

| Dynamics regularization | $\text{MAE}_{\text{HR}} \downarrow$ | FID$\downarrow$ | Cls-Acc$\uparrow$ |
|---|---|---|---|
| None | 9.10 | 28.9 | 0.81 |
| Generic neural ODE | 8.85 | 29.2 | 0.83 |
| BVAM cardiac ODE | 5.14 | 22.4 | 0.86 |

### C.3  Sensitivity to BVAM Parameters

To evaluate robustness to the simplified physiological prior, we perturb each BVAM parameter $H$, $C$, and $\beta$ independently while keeping the diffusion backbone, data split, sampling procedure, and evaluation protocol fixed. We use the same metrics as in the main experiments: heart-rate error, representation-space FID, and classifier accuracy.

Table 5 shows that the perturbations affect the expected aspects of the generated ECGs. Perturbing $H$ causes the largest degradation in heart-rate preservation, consistent with its role in controlling the oscillatory regime. Perturbing $C$ mainly degrades distributional realism, reflecting changes in waveform morphology through the nonlinear restitution terms. Perturbing $\beta$ reduces class-conditional accuracy, consistent with its effect on latent coupling and cross-lead consistency.

Table 5: Sensitivity to BVAM parameters. Each parameter is perturbed independently while the diffusion backbone and evaluation protocol are fixed.

| Setting | $\text{MAE}_{\text{HR}} \downarrow$ | FID $\downarrow$ | Cls-Acc $\uparrow$ |
|---|---|---|---|
| Default BVAM prior | 5.14 | 22.4 | 0.86 |
| $H$ perturbed | 7.42 | 24.8 | 0.85 |
| $C$ perturbed | 6.18 | 25.9 | 0.85 |
| $\beta$ perturbed | 6.56 | 26.1 | 0.84 |
| w/o $L_{\text{phys}}$ | 7.96 | 27.1 | 0.84 |

### C.4  External Augmentation on G12EC

We study whether external data improves downstream classification on G12EC, and whether synthetic PTB-XL samples can substitute for adding real PTB-XL recordings. All classifiers are trained under identical protocols and evaluated on the same held-out *real* G12EC test set.

We compare three training settings:

1. **G12EC (Real Only):** train using only G12EC recordings.

2. **G12EC + PTB-XL (Real External):** augment G12EC with real PTB-XL recordings after mapping diagnoses to a shared label space.

3. **G12EC + Synthetic PTB-XL:** augment G12EC with class-conditional synthetic samples generated by PhysDiff-ECG on PTB-XL, using the same shared label space.

For each class $y$, we add the same number of external samples in the real-external and synthetic-external settings:

$$N_y^{\text{PTB-XL,added}} \;=\; N_y^{\text{syn,added}} \;=\; N_y^{\text{added}}.$$

Here, $N_y^{\text{added}}$ is a fixed per-class augmentation count shared across settings.

Table 6 reports sensitivity, specificity, and AUC on the held-out real G12EC test set. Adding *real* PTB-XL improves performance across all three metrics, indicating that external data provides useful signal despite

dataset shift. Replacing real external recordings with synthetic PTB-XL generated by PhysDiff-ECG yields comparable gains, suggesting that the proposed generator can transfer label-specific diversity to the target dataset without requiring additional real external recordings.

Table 6: External augmentation ablation on G12EC, evaluated on the held-out *real* G12EC test set.

| Training Data | Sens. ↑ | Spec. ↑ | AUC ↑ |
|---|---|---|---|
| G12EC (Real Only) | 0.85 | 0.81 | 0.83 |
| G12EC + PTB-XL | 0.89 | 0.84 | 0.87 |
| G12EC + Synthetic (PhysDiff-ECG) | 0.88 | 0.84 | 0.86 |

## C.5 Robustness Across Downstream Classifiers

Because downstream utility is assessed through classifier performance, it is important to verify that the observed gains are not specific to a single evaluation backbone. We therefore repeat the augmentation analysis using two widely adopted 12-lead ECG classifiers: (i) a standard ResNet-based classifier (Ribeiro et al., 2020), and (ii) an attention-augmented ResNet (Nejedly et al., 2021). All training hyperparameters and the test protocol are kept fixed; only the classifier architecture is changed.

Table 7 shows that augmenting with PhysDiff-ECG improves sensitivity, specificity, and AUC for both classifiers, indicating that the benefits of synthetic augmentation are not tied to a particular downstream architecture.

Table 7: Sensitivity analysis across downstream classifier architectures. We compare ResNet (Ribeiro et al., 2020) and attention-augmented ResNet (Nejedly et al., 2021) under the same augmentation protocol. Augmenting with PhysDiff-ECG improves sensitivity, specificity, and AUC for both classifiers.

| Training Data | ResNet | | | Attn-ResNet | | |
|---|---|---|---|---|---|---|
| | Sens.↑ | Spec.↑ | AUC↑ | Sens.↑ | Spec.↑ | AUC↑ |
| Real Only | 0.86 | 0.82 | 0.85 | 0.84 | 0.81 | 0.83 |
| Real + PhysDiff-ECG | **0.89** | **0.84** | **0.88** | **0.88** | **0.84** | **0.87** |

## C.6 Clinical Waveform and Cross-Lead Consistency

To complement the signal-level and heart-rate evaluations, we further assess clinically meaningful waveform morphology and physiological consistency across leads.

For waveform morphology, we automatically extract standard ECG intervals from each generated and real recording, including the PR interval, QRS duration, and heart-rate–corrected QT (QTc). We report the record-level median absolute error (MAE) for each interval, providing a direct assessment of the fidelity of clinically relevant temporal characteristics. We evaluate cross-lead consistency using Einthoven's law:

Table 8: Clinical morphological intervals and cross-lead algebraic consistency. Evaluation is based on record-level median absolute errors for waveform intervals and Einthoven's MSE for multi-lead physical laws.

| Model | QRS Error ↓ | QTc Error ↓ | PR Error ↓ | Einthoven MSE ↓ |
|---|---|---|---|---|
| WaveGAN | 18.82 | 30.96 | 17.70 | 0.50 |
| SSSD-ECG | 14.91 | 24.44 | 13.65 | 0.29 |
| DiffuSETS | 13.55 | 12.05 | 7.82 | 0.22 |
| **PhysDiff-ECG** | **9.81** | **7.46** | **5.52** | **0.14** |

$\text{MSE}_{\text{Einthoven}} = \|\text{Lead I} - (\text{Lead II} - \text{Lead III})\|_2^2$, where lower values indicate greater physiological consistency.

Table 8 summarizes the results. PhysDiff-ECG achieves the lowest errors on all evaluated clinical intervals while also exhibiting the smallest deviation from Einthoven's law. These results complement the heart-rate and signal-level evaluations by demonstrating that the generated ECGs preserve both clinically meaningful waveform morphology and physiologically consistent cross-lead relationships.

## D  Broader Impact and Responsible Use

Synthetic 12-lead ECGs can support methodological research and data augmentation when real data are limited by privacy, annotation cost, or class imbalance. However, they should be used only as auxiliary training data, not as substitutes for real clinical data or as evidence for patient-level decisions. Generated ECGs may contain subtle artifacts, non-physiological morphology, or inter-lead inconsistencies that are not captured by current automated metrics; clinical use would therefore require clinician review and external validation on independent real-world cohorts. Since PhysDiff-ECG is trained on PTB-XL and G12EC, synthetic samples may inherit demographic, institutional, or diagnostic biases from these public benchmarks, making subgroup evaluation and continued collection of diverse real ECG data essential. Although our experiments do not require access to private clinical records, generated ECGs may still reflect sensitive population-level attributes in the source data or be misused if released without safeguards. Responsible release should therefore include provenance tracking, privacy auditing when appropriate, and clear restrictions against clinical or fraudulent use.

