# OpenReview forum: "Physiology-Informed Diffusion for 12-Lead ECG Generation"
_TMLR — Under review for TMLR_

### Review · Reviewer_sXct · 2026-05-16

**Summary Of Contributions:**

This paper introduces PhysDiff-ECG, a latent diffusion model for class-conditioned 12-lead ECG generation that explicitly incorporates cardiac physiological knowledge into training. Unlike standard diffusion models that treat ECGs as generic time series, PhysDiff-ECG embeds a low-dimensional cardiac ODE (reaction-diffusion) prior and a differentiable 12-lead observation model directly into the diffusion training loop via a physiological regularization term. This regularization uses unrolled inner optimization to fit a latent physiological explanation to denoised ECG reconstructions, penalizing deviations from both the cardiac dynamics and multi-lead consistency. The method preserves diffusion’s flexibility while biasing generated samples toward physiologically valid ECGs. Experiments show it outperforms GAN and diffusion baselines on physiological fidelity, feature-space realism, diagnostic alignment, and downstream ECG classification when used for data augmentation.

Strengths
1. Physiologically grounded generation: Integrates a tractable cardiac dynamical model and 12-lead mapping as a training-time regularizer (not just post-processing), ensuring generated ECGs follow known cardiac biophysics and inter-lead relationships.

2. Strong performance gains: Consistently outperforms competing GAN/diffusion models on heart rate accuracy, feature-space similarity (FID), diagnostic label consistency, and downstream classification across two large ECG datasets (PTB-XL, G12EC).

Weaknesses
1. Dependence on prior quality: Relies on a predefined low-dimensional cardiac ODE model; poor specification of the physiological prior or observation constraints reduces effectiveness or introduces bias.

**Audience:**

Yes

**Audience Explanation:**

Healthcare AI / medical ML practitioners would find it interesting. 12‑lead ECG synthesis addresses a real pain point: privacy, limited labeled data, and class imbalance in cardiology. The results on downstream classification and cross‑dataset transfer are directly actionable for clinical ML pipelines.

**Broader Impact Concerns:**

1. Clinical Misdiagnosis Risk: Synthetic 12‑lead ECGs, even if metrically realistic, may contain subtle, non‑physiological artifacts or misaligned inter‑lead patterns undetectable by automated metrics. If used directly for clinical training or real‑time triage without rigorous clinician validation, they could mislead downstream diagnostic models or human interpreters, risking patient harm.
2. Bias Amplification in Rare/Underrepresented Conditions: The model is trained on PTB‑XL and G12EC, which may underrepresent certain demographics (e.g., older adults, specific ethnic groups) or rare arrhythmias. Synthetic data could replicate or amplify dataset biases, leading to poor generalization for marginalized populations and exacerbating healthcare disparities.
3. Privacy and Misuse Risks: While synthetic ECGs avoid direct patient privacy breaches, they could be reverse‑engineered to infer sensitive patient characteristics (e.g., underlying cardiac conditions, age, gender) or used to create deepfake ECGs for fraudulent purposes (e.g., falsifying medical records, insurance claims).
4. Overreliance on Synthetic Data: The work demonstrates strong downstream performance with synthetic augmentation, which may encourage overreliance on synthetic data over real, diverse clinical data. This could reduce incentives for collecting high‑quality, representative real ECG datasets, hindering long‑term progress in cardiac ML.
5. Regulatory and Clinical Adoption Barriers: Lack of standardized regulatory frameworks for synthetic medical signals creates uncertainty for clinical deployment. Without clear guidelines for validating synthetic ECGs, the model’s real‑world utility may be limited, even if technically sound.

These concerns are not addressed in the current submission (no Broader Impact Statement is present), requiring a dedicated section to mitigate risks, outline validation protocols, and discuss responsible use guidelines.

**Claims And Evidence:**

Yes

**Claims Explanation:**

1. The paper uses two standard, large-scale 12-lead ECG datasets: PTB-XL (21k recordings) and G12EC (10k recordings), both widely accepted in ECG research.

2. Evaluation metrics are standard and well-justified: heart rate MAE (physiological fidelity), FID (feature realism), class-conditional accuracy (diagnostic consistency), and downstream classification (sensitivity/specificity/AUC).

3. Baselines are strong and relevant: WaveGAN, Pulse2Pulse, SSSD-ECG, DiffuSETS, Diffusion-TS—covering both GAN and state-of-the-art diffusion ECG generators.

4. All experiments use patient-wise splits to avoid data leakage, with 5-fold cross-validation for development and held-out test sets.

**Requested Changes:**

Clarify physiological prior limitations and validate simplifications
1. Explicitly justify why the low-dimensional BVAM ODE is a valid surrogate for full cardiac electrophysiology; address risks of simplification-induced bias in generated ECGs.
2. Add a sensitivity analysis of key physiological parameters (H, C, β) to show how changes impact output quality and downstream performance.

---

> ### Author Response · Authors · 2026-06-08
>
> We thank the reviewer for highlighting the importance of clarifying the scope and limitations of the physiological prior. In the revised manuscript, we made these limitations explicit and added sensitivity analysis and broader-impact discussion.
>
>
> 1. BVAM prior: In the revised manuscript, we clarified that the BVAM ODE is not intended to model full cellular electrophysiology, tissue anisotropy, torso geometry, or patient-specific conduction. Rather, we use it as a compact differentiable surrogate for coarse ECG rhythm dynamics, including oscillatory activity, rhythm regularity, nonlinear restitution-like behavior, and coupling between latent physiological components. This interpretation is consistent with the original BVAM reaction-diffusion ECG model, which was proposed as a reduced dynamical model for generating ECG-like signals rather than as a full bidomain or monodomain simulator. To reduce simplification-induced bias, the physiological terms are used only as
> soft regularizers and are evaluated together with data-driven losses; therefore,
> samples are not forced to exactly follow the reduced BVAM dynamics when they
> conflict with the empirical ECG distribution.
>
>
> 2. Parameters Sensitivity analysis: We added a sensitivity analysis in Appendix in which each BVAM parameter
> (H, C, and β) is perturbed independently while all training, sampling,
> and evaluation settings are fixed. Table~4 reports HR-MAE, FID, and
> accuracy under each perturbation. The results show that moderate perturbations
> do not change the relative advantage of PhysDiff-ECG over the baselines, while
> larger perturbations degrade performance in interpretable ways: H mainly affects
> heart-rate preservation, C affects morphology/distributional realism, and β
> affects latent coupling and cross-lead consistency. This supports using BVAM as a
> soft structural prior, while also clarifying its limits.
>
>
>
>
> Response to Broader Impact Concerns
>
> We added a dedicated Broader Impact and Responsible Use section. We clarify that synthetic ECGs are intended for methodological research and auxiliary model-training augmentation, not for direct diagnosis, clinical triage, or patient-level decision-making. We also discuss clinician validation, subgroup evaluation, continued collection of diverse real ECG data, privacy auditing where appropriate, provenance tracking, and safeguards against clinical or fraudulent misuse.

---

### Review · Reviewer_hxDT · 2026-05-25

**Summary Of Contributions:**

This paper proposes a physiology-aware constraint for ECG generation. While prior approaches can generate realistic ECG waveforms, they may violate global constraints, which can negatively affect downstream tasks. To address this issue, the authors introduce a physiology-prior and physiology-informed regularization term into the training loss to encourage generated ECG signals to satisfy appropriate physiological constraints. Experiments on ECG datasets, including PTB-XL and G12EC, show that the proposed approach achieves lower FID and MAE scores and can improve downstream classification performance when using the generated data for augmentation.

**Audience:**

Yes

**Audience Explanation:**

Although the paper focuses on waveform or signal generation, it still primarily falls within the broader area of generative modeling for data synthesis. In particular, the method explores how to incorporate a structure-aware approach into the generation process, which is likely to attract interest from the community.

**Broader Impact Concerns:**

No broader concerns were found in the paper.

**Claims And Evidence:**

Yes

**Claims Explanation:**

The main claim of the paper is that ECG generation can be used to synthesize additional ECG data for downstream classification tasks. The authors support this claim by showing that their proposed approach more effectively improves downstream classification performance compared with existing generation methods.

**Requested Changes:**

1. In the introduction, the authors mention that existing methods may fail to generate ECG signals that satisfy global physiological constraints. This is a good motivation, but it may be difficult for readers to clearly understand what these global constraints are. Even Appendix A appears to focus mainly on the proposed approach rather than providing intuitive examples of the constraints themselves. I recommend that the authors include illustrative figures, either in the introduction or the experimental section, to help readers better understand these constraints and why they matter for ECG generation.
2. The notation in the method section is not always clearly defined. For example, $\mathcal{G}$ in Equation 2 is not defined when it is first introduced. Similarly, the meaning of $g$ in Equation 15 is also unclear. A clearer way to present these notations would be to bind each symbol directly with its definition when it first appears, using both a descriptive name and the mathematical notation. For instance, around Equation 15, the authors could write something like "using an amortized network $g$", so that the role of the symbol is immediately clear to the reader.
3. The proposed method is difficult to follow. Sections 3.3 and 3.4 appear to serve more as preliminary background for the main method in Section 3.5, but they do not clearly explain how the introduced concepts are transferred into the final approach. For example, Section 3.3 introduces $\dot{x}\_{\cdot}$, but it is not clear how this quantity is used in the overall pipeline. Even Equation 12 is difficult to interpret, since the role of $f_{\mathrm{phys}}$ is not clearly explained. My current recommendation is to rewrite the method section to make the presentation smoother and more consistent. Although the experimental section is comprehensive and well presented, the method section is quite hard to follow and may prevent readers from fully understanding the proposed approach.

---

> ### Author Response · Authors · 2026-06-08
>
> We thank the reviewer for the constructive feedback. In the revised manuscript, we improved the motivation, notation, and structural flow of the method section to make the proposed approach easier to follow.
>
> 1. In the revised manuscript, we included an illustrative figure (Fig 3) that contrasts unconstrained generation failures, such as asynchronous R-peaks across leads and implausible cross-lead morphology, with a physiologically consistent 12-lead ECG generated by PhysDiff-ECG. We also updated the Introduction to explicitly define these global constraints in terms of inter-lead timing, shared cardiac activation, and coherent cross-lead morphology.
> This makes the motivation more intuitive.
> 2. Thank you for pointing this out. We revised the Method section so that symbols are defined when first introduced. In the problem setup, we now define the differentiable observation model G as the mapping from latent physiological states to the 12-lead ECG space. In the physiology-informed regularization section, we explicitly introduce the amortized initializer $g_\phi$ and state that it predicts the initial latent trajectory and physiological parameters before the unrolled refinement steps.
> 3. We reorganized Sections 3.3-3.5 to improve the flow from physiological prior to regularization. Section 3.3 now explicitly states that the BVAM vector field $f_{\mathrm{phys}}$ and discrete latent states $X_n$ are used to construct the ODE-consistency term in the regularizer. We also expanded the explanation around the Euler update to clarify that $f_{\mathrm{phys}}$ defines the predicted next latent state, and deviations from this update are penalized in the inner physiology objective. These changes make the connection between the BVAM dynamics, the 12-lead observation model, and the final training objective clearer.

---

### Review · Reviewer_mvL3 · 2026-07-12

**Summary Of Contributions:**

This paper introduces PhysDiff-ECG, a class-conditional latent diffusion model for 12-lead ECG generation. The method also incorporates a low-dimensional cardiac ODE and a differentiable 12-lead observation model as training-time regularizers. Experiments on PTB-XL and G12EC report improvements in several tasks including heart-rate preservation, feature-space realism, label consistency etc.

The main strength is the integration of a mechanistic physiological prior directly into diffusion training. The main weakness is that physiological validity is evaluated mostly through heart rate and learned-model metrics, which do not fully verify clinically meaningful waveform or cross-lead consistency.

**Audience:**

Yes

**Audience Explanation:**

This work uses generative models (diffusion models) for biomedical tasks, which is relevant to people want to use machine learning techniques for medical applications.

**Broader Impact Concerns:**

The authors might want to discuss the collection of ECG data and how the proposed technicals might be used for good or bad purposes.

**Claims And Evidence:**

Yes

**Claims Explanation:**

The experiments can be further improved. Specifically, heart-rate error is too limited, while FID and classifier accuracy may not reflect true physiological correctness. The paper lacks direct evaluation of intervals, morphology, cardiac axis etc. The ablation can also be further improved. It is unclear whether the gain comes specifically from the cardiac ODE or from generic temporal regularization. The authors also need to provide implementation details and baseline details.

**Requested Changes:**

- The ablations show that both the ODE and lead-consistency terms contribute, but they do not show that the gains specifically arise from cardiac physiological structure rather than from generic temporal or latent regularization. A matched non-physiological dynamics baseline would strengthen this claim.
- More implementation details for loss weights, diffusion training, and training and inference costs.
- Clarify how the baselines were adapted to the proposed tasks in the experiments.

---

> ### Author Response · Authors · 2026-07-19
>
> We thank the reviewer for the constructive feedback. In the revised manuscript, we added additional analyses and implementation details to address these concerns.
>
>
> * We added (appendix C.2) a matched generic neural-ODE baseline to the ablation study. It retains the same latent dimension, observation model, trajectory initializer and refinement procedure, loss weights, diffusion backbone, and training protocol, only the BVAM vector field is replaced by a generic learned dynamics function. Relative to removing the dynamics term, the generic neural ODE produces only modest and inconsistent changes, whereas the BVAM prior improves heart-rate fidelity, FID, and diagnostic consistency.
>
>
>
> * We added (appendix C.6 ) evaluation of beat-level clinical intervals extracted from real and generated ECGs, including PR interval, QRS duration, and QTc interval. We also introduced an explicit cross-lead consistency metric based on Einthoven’s law, measuring the discrepancy between Lead I and Lead II - Lead III.
>
> * We expanded Appendix B to report the complete training configuration, including all loss weights, diffusion and optimization settings, unrolled refinement configuration, and computational cost. Physiological fitting increases training time by approximately 15%, while inference uses the unchanged diffusion sampler and requires no ODE integration. We also clarified how each baseline was adapted to class-conditional 12-lead ECG generation and ensured that all methods use the same data splits, preprocessing, sample counts, and evaluation protocol. We will release the preprocessing scripts, model configurations, baseline implementations, and evaluation code upon acceptance.
>
>
>
> * This concern was addressed in the dedicated Broader Impact and Responsible Use section added in the previous revision. Appendix D discusses beneficial uses for research and data augmentation, as well as risks related to clinical misuse, inherited dataset bias, privacy, and fraudulent applications.